# Effects of Ultra-High-Pressure Treatment on Chemical Composition and Biological Activities of Free, Esterified and Bound Phenolics from *Phyllanthus emblica* L. Fruits

**DOI:** 10.3390/molecules29133181

**Published:** 2024-07-03

**Authors:** Taiming Chen, Shuyue He, Jing Zhang, Huangxin Wang, Yiqing Jia, Yaping Liu, Mingjun Xie, Guiguang Cheng

**Affiliations:** 1Faculty of Food Science and Engineering, Kunming University of Science and Technology, Kunming 650500, China; ming19970515@163.com (T.C.); z1241742925@163.com (J.Z.); whx9806@126.com (H.W.); jiaq7207@163.com (Y.J.); liuyaping@kust.edu.cn (Y.L.); 2Linyi Technician Institute, Linyi 276005, China; kmlgdx303@163.com; 3Faculty of Environmental Science and Engineering, Kunming University of Science and Technology, Kunming 650500, China

**Keywords:** *Phyllanthus emblica* L. fruits, phenolics, antioxidant, cytoprotective activity, ultra-high-pressure treatment, UHPLC-ESI-HRMS/MS

## Abstract

*Phyllanthus emblica* L. fruits (PEFs) were processed by ultra-pressure (UHP) treatment and then extracted by the ultrasonic-assisted extraction method. The influence of UHP on the phenolic composition, enzyme inhibitory activity and antioxidant activity of the free, esterified, and bound phenolic fractions from PEFs were compared. UHP pretreatment of PEFs significantly increased the total phenolic and flavonoid contents (*p* < 0.05). A total of 24 chemical compositions were characterized in normal and UHP-treated PEFs by UHPLC-ESI-HRMS/MS. Compared with normal PEFs, these three different phenolic fractions had stronger antioxidant activities and inhibitory effects on the intracellular reactive oxygen species (ROS) production in H_2_O_2_-induced HepG2 cells (*p* < 0.05). The ROS inhibition might be due to an up-regulation of the expressions of superoxide dismutase (SOD) and glutathione (GSH) activities. In addition, these three different phenolic fractions also significantly inhibited the activities of metabolic enzymes, including α-glucosidase, α-amylase and pancreatic lipase. This work may provide some insights into the potential economics and applications of PEFs in food and nutraceutical industries.

## 1. Introduction

*Phyllanthus emblica* L., belonging to the Euphorbiaceae family, is a perennial plant that is widely distributed in China, India, Thailand, Malaysia and other countries in Southeast Asia [1]. It is an important economic crop in China [2]. Its fruits have been widely processed as juice, beverages, jam, and cosmetics due to their special taste, nutrients and bioactive phytochemicals [3]. In addition, the fruits and roots have traditionally been used as herbal medicine for treating eczema, cough and sore throat in China [4]. Many studies reported that *P. emblica* fruits (PEFs) had significant anti-inflammatory, antibacterial, neuroprotective, hepatoprotective, antihyperglycemic and antihyperlipidemic effects [5,6,7,8]. In addition, phytochemical investigations of PEFs reported a series of phenolic acids, flavonoids, and tannins, and these phenolic compounds exist in bound form [9].

Phenolic compounds can be classified into the free, esterified and bound forms based on their extractability and interaction with cell wall components [5]. The free and esterified phenolics localized in the vacuoles of plant cells could be easily extracted by different organic solvents [10]. The bound phenolics mostly bind to macromolecules in the cell wall matrix, such as cellulose, hemicellulose, lignin, pectin and proteins, thereby the extraction yield is low. Therefore, the research on the chemical composition and biological activity of bound phenolics is limited [11]. In the past decade, the issue of how to improve the extract yield and illustrate the chemical constituents of bound phenolics has attracted increasing attention.

Recently, many modern non-thermal techniques, such as ultra-high-pressure homogenization, thin-film short-wave ultraviolet radiation, and high hydrostatic pressure, have been established to improve the potential in terms of the bioaccessibility and bioavailability of bound phenolics [12,13,14]. Ultra-high hydrostatic pressure (UHP) could damage the cell walls and the chemical hydrogen, ester, disulfide and carbon–carbon bonds between the bound components and the plant cell matrix. Some studies had found evidence that UHP treatment could increase the extraction yield of polysaccharides [15], flavonoids [16] and carotenoids [17] in a short time and at a low temperature [18].

In this study, the UHP processing method were used to extract the free, esterified, and bound phenolics in PEFs. Furthermore, the chemical compositions and the antioxidant activities, cytoprotective effects and enzyme inhibitory activities of different phenolic fractions were determined.

## 2. Results and Discussion

### 2.1. Characterization of Phenolic Compounds in Different Fractions of PEFs

The phenolic compounds in the free, esterified and bound forms of PEFs were analyzed by ultra-high-performance liquid chromatography–high-resolution mass spectrometry (UHPLC-HRMS/MS). The total ion current chromatograms of the three different phenolic forms are presented in Figure 1.

The characterization of each compound was performed by comparing the mass-related data with those of the corresponding authentic standards or with mass information available in the literature or the Human Metabolome Database (https://hmdb.ca/structures/search/metabolites/structure, accessed on 31 May 2024). A total of 24 constituents were identified. The distribution of the chemical components in each group is shown in Figure 2. 

The specific information, including the retention time, molecular formula, mass and MS/MS ion data, and delta ppm, are summarized in Table 1.

#### 2.1.1. Phenolic Acids

Mucic acid (**1**) was detected with [M-H]^-^ ion at *m/z* 209.0302, which was consistent with the MS/MS of the standard. Chebulic acid (**2**) was identified based on its fragmentation pattern with ions at *m/z* 355.0317 [M-H]^-^ and 336.7523 [M-H-H_2_O]^-^ [6,30]. Compound **3** was assigned the molecular formula of C_13_H_16_O_10_ by the [M-H]^-^ ion at *m/z* 331.0680. The neutral loss of a 162 Da part (C_6_H_10_O_5_) corresponding to a glucosyl moiety was readily observed in the MS^2^ experiment. Thus, this peak was determined to be gallic acid glycoside [19], and the fragment pathway is exhibited in Figure 3a. 

Galic acid (**4**) had an [M-H]^-^ ion at *m/z* 169.0137 and an [M-COOH]^-^ ion at *m/z* 125.0333 [6]. Compounds **5** and **14** also exhibited a same fragment ion at *m/z* 169.01, which was characterized as a gallic acid derivative [31]. Methyl gallate (**6**) had an [M-H]^-^ ion at *m/z* 183.0295 (C_8_H_7_O_5_^−^), and a fragment ion at *m/z* 169.0124 was yielded via successive elimination of the CH_2_ (14 Da) moiety from the precursor ion, and the fragment ion *m/z* 124.0153 was observed, consistent with the standard data [32]. Compound **7** was identified as digalloylglucose based on the [M-H]^-^ ion at *m/z* 453.0793, with the molecular formula of C_20_H_20_O_11_ determined by losing a galloyl (152 DA) unit to produce an MS/MS fragment ion at *m/z* 331.0574 [6]. The possible fragment pathway of compound **7** is exhibited in Figure 3b. Digallic acid (**8**) was identified based on an [M-H]^-^ ion at *m/z* 321.2106 (C_14_H_10_O_9_) with an MS/MS fragment ion at *m/z* 125.0235 [M-H-196]^-^ [33]. The [M-H]^-^ ion at *m/z* 291.015 and the MS/MS fragment ions at *m/z* 247.0253 [M-H-COOH]^-^ and 191.0347 (C_10_H_7_O_4_) indicated that compound **9** was brevifolin carboxylic acid [20]. Gallic acid ethyl ester (**13**) was identified by the typical loss of CO_2_ [M-H-44 DA]^-^ [22]. Ellagic acid (**15**) had a precursor ion [M-H]^-^ ion at *m/z* 300.9971 with the molecular formula of C_14_H_6_O_8._ The fragment ions at *m/z* 283.9963 and *m/z* 257.0099 were yielded by the loss of the H_2_O (18 Da) unit and CO_2_ (44 Da). Hence, compound **15** was further identified as ellagic acid by comparison with its commercial standard [25]. Azelaic acid (**17**) had an [M-H]^-^ ion at *m/z* 187.0973 and an MS/MS fragment ion at *m/z* 169.0864 [M-H-H_2_O]^-^ [26].

#### 2.1.2. Tannins

Galloyl-HHDP-glucose (**11**) had an [M-H]^-^ ion at *m/z* 633.0775 (C_27_H_22_O_18_) and MS/MS fragment ions at *m/z* 463.1024 [M-H-galloyl-H_2_O]^-^ and 301.0135 [M-H-galloyl-H_2_O-Hex]^-^, which was inconsistent with the previously reported data [6,21]. Compounds **10** and **12** had the same [M-H]^-^ ions at *m/z* 635.091 and MS/MS fragment ions at *m/z* 313.0574 [M-H-galloyl-galloyl]^-^, which was consistent with the data reported in the literature [6]. The possible fragment pathway is exhibited in Figure 3c.

#### 2.1.3. Flavonoids

Tricetin (**18**) was identified by comparison with the data reported in the literature [25]. Quercetin (**20**) had an [M-H]^-^ ion at *m/z* 301.2030 with the molecular formula of C_15_H_9_O_7_. It was further identified by comparison to the retention time of the standard. Quercitrin (**16**) exhibited an [M-H]^-^ ion at *m/z* 447.0926 with the molecular formula of C_21_H_20_O_11_, and it had a fragment ion at *m/z* 283.1923 for [M-2H-162] [34] (Figure 3d). Compound **22** displayed an [M-H]^-^ ion at *m/z* 285.2080 with the molecular formula of C_15_H_10_O_6_, and it had a main MS/MS fragment ion at *m/z* 153.0175. Thus, compound **22** was identified as kaempferol [6,24].

#### 2.1.4. Others

Trihydroxy-10-trans-octadecenoic acid (**23**) displayed an [M-H]^-^ ion at *m/z* 329.2343, with the molecular formula C_18_H_34_O_5_, and MS/MS fragment ions at 171.1018 and 211.1348, which was consistent with the literature [28]. Compound **26** displayed an [M-H]^-^ion at *m/z* 277.1452, in agreement with the molecular formula C_16_H_22_O_4_, which was identified as mono-octyl phthalate [29]. Hydroperoxy-octadecadienoic acid (**27**) was identified by the [M-H]^-^ ion at *m/z* 311.2283 (C_8_H_32_O_4_) and the MS/MS fragment ions at *m/z* 183.0118 and 311.169 [28]. Compounds **24** and **25** had the same [M-H]^-^ ion at *m/z* 433.260, indicating that the molecular formula was C_25_H_38_O_6_. Compound **19** exhibited an [M-H]^-^ ion at *m/z* 399.0943, and the molecular formula was predicted to be C_17_H_18_O_11_.

### 2.2. Determination of Total Phenolic, Total Flavonoid and Total Tannin Contents

The total phenolic content (TPC), total flavonoid content (TFC) and total tannin content (TTC) of the three phenolic fractions from normal and UHP-treated PEFs were determined.

As shown in Figure 4, the normal and UHP-treated PEFs were rich in phenolics, especially free phenolics, which was consistent with a previous report [35]. Among the three phenolic fractions of normal PEFs, the esterified phenolic fraction (EP) possessed the highest TPC, TFC, and TTC values, with 423.12 ± 12.63 mg GAE/g extract, 164.76 ± 7.89 mg RE/g extract and 164.76 ± 7.89 mg RE/g extract (*p* < 0.05), followed by the free phenolic fraction (FP). The bound phenolic fraction (BP) had the lowest TPC with 223.12 ± 10.63 mg GAE/g extract, the lowest TFC with 134.76 ± 7.89 mg RE/g extract and the lowest TTC with 24.76 ± 8.89 mg RE/g extract (*p* < 0.05), respectively. The UHP preprocessing treatment significantly enhanced the TPC values in the FP and BP fractions to 1.30 and 1.34 times higher than those in the normal PEFs (*p* < 0.05). After UHP treatment, the TFC values increased approximately 1.25, 1.86, and 1.34 times. For the tannins, the variation trend was similar to the phenolics, where the TTC values in the FP and BP fractions were 7.0 and 3.80 times increased compared to those in the normal PEFs, except for the EP fraction, respectively.

According to the TPC, TFC and TTC results of the different phenolic fractions, UHP treatment is an effective method for enhancing the extractive phenolic compounds in PEFs, especially for the flavonoids and tannins [36]. It is possible that some esterified phenolic fractions are converted into free phenolics due to the ultra-high-pressure pretreatment. In addition, the bonds between the hydroxyl groups in the flavonoids and tannins and the cell wall components change the membrane permeability, and some phenolics and flavonoids in the membrane are easy to extract in the subsequent process [37].

### 2.3. Antioxidant Activities

In this study, the antioxidant activities of all the phenolic fractions from normal and UHP-treated PEFs were evaluated in terms of the scavenging abilities of 2,2-diphenyl-1-picryhydrazyl (DPPH) and 2,2′-azino-bis (3-ethylbenzothiazoline-6-sulfonic acid) (ABTS), as were the ferric-reducing antioxidant power (FRAP), the inhibitory effect on intracellular reactive oxygen species (ROS) generation and the intracellular antioxidant enzyme activities.

#### 2.3.1. DPPH Radical-Scavenging Activity

As shown in Figure 5a, all the phenolic fractions from the normal PEFs exhibited a high DPPH radical-scavenging capacity with a dose-dependent relationship. These three phenolic fractions presented numerous phytoconstituents, such as gallic acid, ellagic acid and different tannins [1]. The median inhibitory concentration (IC_50_) values of the esterified, free and bound phenolic fractions of normal PEFs were 15.71 ± 1.01, 18.81 ± 0.89, and 22.14 ± 0.98 μg/mL, respectively. The esterified fraction had the strongest DPPH radical-scavenging activity in comparison with the FP and BP (*p* < 0.05). After UHP treatment, the IC_50_ values of the free, esterified and bound phenolic fractions were 5.14 ± 0.71, 2.52 ± 0.56, and 11.38 ± 0.49 μg/mL, respectively. Similarly, the free fraction had the strongest DPPH radical-scavenging activity among the three different fractions. The IC_50_ values of all the fractions clearly suggested that UHP treatment could dramatically enhance the DPPH radical-scavenging activities of the free, esterified, and bound phenolic fractions of PEFs (*p* < 0.05). This may be related to the increase in the phenolic numbers and contents after UHP treatment. Moreover, based on the TFC values and DPPH radical-scavenging activities of the six fractions from the normal and UHP-treated PEFs, a positive correlation can be observed (r = 0.702, *p* < 0.05). These findings revealed that the TPC or TFC values are positively related to the antioxidant capacity. This finding is consistent with the results of previous studies [36].

#### 2.3.2. ABTS Radical-Scavenging Capacity

The ABTS radical-scavenging capacities of all the tested samples are illustrated in Figure 5b. All the fractions exhibited good ABTS radical-scavenging capacities with a dose-dependent relationship. The IC_50_ values of the esterified, free, and bound phenolic fractions from normal PEFs were 9.46 ± 0.48, 23.30 ± 0.58, and 26.07 ± 0.29 μg/mL, respectively. The esterified phenolic fraction also exhibited the strongest ABTS radical-scavenging capacity, followed by the free phenolic fraction. The esterified phenolic fraction contained a large amount of gallic acid, which has a very strong antioxidant capacity [38]. The IC_50_ values of the esterified, free, and bound phenolic fractions from the UHP-treated PEFs were 15.99 ± 0. 65, 12.40 ± 0.47, and 24.43 ± 0.13 μg/mL, respectively. Thus, UHP treatment enhanced the ABTS radical-scavenging capacities of two different phenolic fractions from PEFs. Similar to the findings of the DPPH radical-scavenging assay, the ABTS radical-scavenging activity had positive correlations with the TPC, TFC and TTC values.

#### 2.3.3. FRAP Evaluation

The FRAP values of the free, esterified, and bound phenolic fractions extracted from the non-treated and UHP-treated PEFs are presented in Figure 5c. The FRAP values of all the samples increased with increasing concentrations. The antioxidant properties were measured by the ferric-reducing antioxidant power (FRAP) [39], and the result showed that the esterified phenolic fraction from normal samples had the highest FRAP value, followed by free phenolic fraction, whereas the bound phenolic fraction possessed the lowest FRAP value. In addition, the FRAP values of the esterified, free, and bound phenolic fractions from the UHP-treated PEFs were all significantly higher than those of their counterparts from the normal PEFs at all the tested concentrations (*p* < 0.05).

### 2.4. Protective Effect on H_2_O_2_-Induced Oxidative Stress in HepG2 Cells In Vitro

Phenolics can protect cells from oxidative damage by inhibiting the production of excessive ROS [40]. In this study, tests to determine whether the different phenolic fractions of PEFs had the capacity to inhibit ROS generation were performed in H_2_O_2_-induced HepG2 cells.

#### 2.4.1. Cytotoxic Activities

The HepG2 cell viabilities were measured after co-incubation with the different phenolic fractions from PEFs with or without UHP pretreatment by the methylthiazol-2-yl-2,5-diphenyl tetrazolium bromide (MTT) method [41]. As shown in Figure 6, the cell viability after all the samples were treated was greater than 90%, and there was no significant difference compared to the control group. Hence, all of the samples were non-toxic to HepG2 cells at the concentration of 100 μg/mL.

#### 2.4.2. Effect of Different Phenolic Fractions from PEFs on Intracellular Antioxidant Enzyme Activities

The endogenous antioxidant enzyme systems, including superoxide dismutase (SOD) and glutathione (GSH), play an indispensable role in preventing oxidative stress-related damage [40,41]. SOD is an active substance derived from organisms and plays an important role in the active oxygen-scavenging system in organisms [42]. SOD can specifically remove harmful free radicals from the body and reduce the damage to the body caused by free radical oxidation [43].

As shown in Figure 7, the activities of SOD and GSH significantly decreased in the H_2_O_2_-treated group when compared to those in the control group (*p* < 0.05). Figure 7b illustrates that UHP treatment significantly increased the SOD activity of all the different phenolic fractions of the PEFs. There was a significant difference between the SOD activity of the normal and UHP-treated PEFs. Among the three different phenolic fractions obtained from the normal and UHP-treated PEFs, the FP exhibited the strongest improvement activity in terms of the SOD expression. By comparison with the control group, the FP increased by 49%, followed by the EP (*p* < 0.05).

Reduced glutathione (GSH) is a low-molecular-weight scavenger with the physiological function of removing O_2_^-^, H_2_O_2_ and LOOH [44]. As shown in Figure 7a, the GSH activity increased by 32%, 34% and 5%, respectively, in the UHP-treated PEFs compared with the normal PEFs. The GSH activity of the three phenolic fractions showed significant differences with the control group (*p* < 0.05). The difference in GSH activity between the free and bound phenolics was not significant, but there were significant differences between the three fractions and the control group (*p* < 0.05). The mechanism may be related to the fact that phenolics could directly scavenge free radicals or that they increase GSH levels by promoting the synthesis of glutathione synthase.

#### 2.4.3. Inhibitory Effect of Different Phenolic Fractions from PEFs on Intracellular ROS Generation

As shown in Figure 7d, the relative amount of intracellular ROS significantly increased to 212.33 ± 10.67% after the H_2_O_2_ treatment when compared with the control group. After the treatment of the six fractions from the PEFs with or without UHP pretreatment, the ROS production effectively decreased in a concentration-dependent manner (*p* < 0.05). The FP, EP, and BP from the UHP-treated PEFs had lower values than those from the normal PEFs (*p* < 0.05), indicating that UHP treatment could effectively enhance the inhibitory effect of the three phenolic fractions on intracellular ROS generation. Additionally, the FP exhibited the strongest intracellular ROS-scavenging activity, whereas the BP showed the weakest intracellular ROS inhibitory effect in comparison with the other extracts (Figure 7c). The inhibitory effect of the three phenolic components on intracellular ROS production might be related to the chemical constituents and contents of phenolics, flavonoids and tannins [45]. In addition, the synergistic effects between different phenolic compounds might be another influencing factor [46].

### 2.5. Enzyme Inhibitory Activity of Different Phenolic Fractions from PEFs

Both α-glycosidase and α-amylase inhibitors are used to prevent diseases such as obesity, hyperglycemia and diabetes. The inhibitory effects of these phenolic fractions against α-glucosidase and α-amylase were determined and the results are described in Table 2. All the phenolic fractions showed significant inhibitory capacity in relation to α-glucosidase and α-amylase (*p* < 0.05). The IC_50_ values of the esterified, free, and bound phenolic fractions from normal PEFs on α-glycosidase were 312.4 ± 2.21, 19.6 ± 1.01, and 87.65 ± 3.42 μg/mL, respectively. The free phenolic fraction exhibited the strongest inhibitory effect, followed by the bound phenolic fraction, whereas the esterified phenolic fraction had the lowest activity (*p* < 0.05). The findings revealed that the phenolic compounds had higher IC_50_ values than that of acarbose, indicating that the phenolic compounds had a strong hypoglycemic effect. The pancreatic lipase inhibitory effects of all the tested samples are illustrated in Table 2.

The results showed that all the phenolic fractions from the PEFs, regardless of UHP treatment, exhibited good inhibitory effects on lipase activity with a dose-dependent relationship at the tested concentrations. The IC_50_ values of the esterified, free, and bound phenolic fractions of normal PEFs were 241.25 ± 8.21, 62.92 ± 2.34, and 613.89 ± 9.83 μg/mL, respectively. The free fraction also exhibited the highest good inhibitory effects on lipase activity (*p* < 0.05), followed by the esterified fraction. Interestingly, the IC_50_ of the bound phenolic fraction from the UHP-treated PEFs is close to that of the positive control, orlistat. The results showed that the phenolic compounds contributed significantly to the inhibitory effect on lipase.

### 2.6. Multivariate Analysis

The total variation explained was 74.30% with PC1 (51.66%) and PC2 (22.64%) (Figure 8). PC1 was more closely related to the SOD, GSH enzyme activity, TFC, TPC and TTC. The scores showed that the UFP-treated PEFs had a higher material content and improved biological activity. Additionally, PC2 was more closely relate to α-glucosidase, α-amylase, and the ROS inhibition ratio. The UFP and UBP groups showed a negative correlation with PC2, indicating that the UFP and UBP PEFs had weak hypoglycemic activity. However, the EP group showed significant hypoglycemic and inhibition of ROS production activity.

### 2.7. Correlation Network

The relationship (positive correlation) between the chemical composition and the biological activity of PEFs is shown in Figure 9. The scores of the Edge showed that gallic acid and its derivatives had a significant influence on the biological activity (Appendix A). The ABTS-scavenging activity was closely related to kaempferol (0.65), azelaic acid deoxyhexose (0.85), dihydroxyhexadecanoic acid (0.85), brevifolin carboxylic acid (0.70) and gallic acid derivatives [47,48]. The SOD and GSH were more affected by TPC. The UHP-treated PEFs showed a higher total flavonoid content and also had better potential to increase the enzyme activities of SOD and GSH. Gallic acid had a great influence on ROS production (0.60). These results are consistent with previous reports [49,50]. There was a positive relationship between the TFC value and the inhibition rate of α-amylase activity (0.71), which was also confirmed by the IC_50_ value of the UFP (1.74) [51].

## 3. Material and Methods

### 3.1. Chemical and Reagents

LC/MS-grade acetonitrile was purchased from Merck (Darmstadt, Germany). Ultra-pure water was deionized using a Milli-Q system (Millipore, Bedford, MA, USA). Folin–Ciocalteu reagent, 1,3,5-tri(2-pyridyl)-2,4,6-triazine (TPTZ), 2,2-diphenyl-1-picryhydrazyl (DPPH), 2,2′-azino-bis (3-ethylbenzothiazoline-6-sulfonic acid) (ABTS), methylthiazol-2-yl-2,5-diphenyl tetrazolium bromide (MTT), and 2′,7′-dichlorofluorescin diacetate (DCFH-DA) were purchased from Sigma-Aldrich (Shanghai, China). Superoxide dismutase (SOD), and glutathione (GSH) were purchased from Sigma-Aldrich (Shanghai, China). Fetal bovine serum (FBS), streptomycin penicillin and Dulbecco’s modified Eagle’s medium (DMEM) were obtained from Gibco (Grand Island, NY, USA). The other organic solvents of analytical grade were purchased from Tianjin Fengchuan Chemical Reagent Co., Ltd. (Tianjin, China). 

### 3.2. Preliminary Treatment

Fresh PEFs were obtained from Pu’er city in the Yunnan Province of China in July 2017 (22°83′42″ N, 100°99′05″ E). The PEFs were manually picked and then dried using a vacuum freeze-dryer (Alpha 1–2 LD plus, Christ, Osterode, Germany). Thereafter, the powdered PEFs (100.0 g of each group) were treated by UHP equipment (HHP-600, Baotou KeFa High Pressure Technology Co., Ltd., Baotou, China). The UHP treatment of the PEFs was performed at 400 MPa for 10 min based on the result of our preliminary experiment.

### 3.3. Extraction of Different Phenolic Fractions from PEFs

The extraction methods for the free, esterified, and bound phenolics were carried out according to a reported method with slight modifications [36]. The sample with or without UHP treatment was firstly degreased with n-hexane at a ratio of 1:5 by the Soxhlet extraction method. After filtration with filter paper, the degreased sample was obtained and extracted by 70% methanol at the ratio of material to liquid of 1:10 for 30 min by ultrasound-assisted extraction (n = 3). The supernatant was collected by centrifugation (1200× *g* 10 min 25 °C) and evaporated in a vacuum using a rotary evaporator (45 °C) to yield a crude extract (CE). Then, the CE was processed for further fractionation of the free, esterified, and insoluble phenolic metabolites.

For the free phenolic fraction, the CE was firstly dissolved with 6 M hydrochloric acid to obtain an acidified aqueous solution with a pH of 2, and then fractioned with ethyl ether (1:1 *v*/*v*) three times. Furthermore, the ethyl acetate extraction solution was combined and evaporated using a rotary evaporator to obtain the free phenolic fraction. For the esterified phenolic fraction, the aqueous phase was hydrolyzed with 4 M NaOH (1:1 *v*/*v*) for 4 h in the aqueous phase at room temperature and adjusted to a pH of 2 by 6 M hydrochloric acid. Then, the acid-aqueous phase was extracted with ethyl acetate (1:1 *v*/*v*). The ethyl acetate layer was collected and evaporated in a vacuum to obtain esterified phenolic metabolites. For the extraction of the bound phenolic fraction, after the extraction of the free phenolic fraction and esterified phenolic fraction, the remaining solid residue was mixed with 4 M NaOH at a ratio of 1:10 (*w*/*v*) in a shaking water bath for hydrolysis for 4 h at room temperature. Then, the hydrolysate was adjusted to a pH of 2 and extracted with ethyl acetate (1:1 *v*/*v*). The combined ethyl acetate layer was evaporated in a vacuum to obtain the insoluble bound phenolic fraction.

### 3.4. Determination of Total Phenolics, Flavonoids and Tannins Contents

The total phenolic content (TPC) of the free, esterified, bound phenolic metabolites in the normal and UHP-treated samples was determined according to our previously described Folin–Ciocalteu method [52]. The absorbance was measured at 765 nm by a Spectra Max M5 microplate reader, and the TPC was expressed as milligrams of gallic acid equivalent (GAE) per gram of extract. The total flavonoids content (TFC) was determined by the aluminum chloride method [18], and expressed as milligrams of rutin equivalent (RE) per gram of extract. The total tannins content was measured by modified phosphomolybdium tungstic acid-casein [53]. The results of the TTC were expressed as milligrams of rutin equivalent (RE) per gram of extract.

### 3.5. Identification of Phenolics by UHPLC-ESI-HRMS/MS

The chemical constituents of the free, esterified, bound phenolic metabolites in the normal and UHP-treated samples were analyzed by ultra-performance liquid chromatography (Thermo Fisher Scientific, Bremen, Germany) on an Agilent C_18_ column (2.1 × 100 mm, 1.9 μm, Agilent, Santa Clara, CA, USA) equipped with a Q-Exactive Orbitrap mass spectrometer (Thermo Fisher Scientific, Bremen, Germany). The chromatographic separation was optimized with a good baseline and resolution. The mobile phases were acidified water (A, 0.1% formic acid) and acetonitrile (0.1% formic acid, solvent B). A gradient program was carried out as follows: 0–5 min, 5% B; 5–10 min, 5%–40% B 10–20 min, 40%–60% B; 20–30 min, 60%–100% B; 30–32 min, 100% B; 32.01–35 min, 5% B. The flow rate was 0.2 mL/min, the injection volume was 2 μL, and the column oven temperature was maintained at 35 °C. The HR-ESI-MS/MS data were analyzed on a Q-Exactive Orbitrap mass spectrometer (Thermo Fisher Scientific, Bremen, Germany). The specific mass parameters were as follows: resolution, 70,000; auxiliary gas flow, 8 L/min; sheath gas flow rate, 32 L/min; sweep gas, 4 L/min; S-lens RF level, 50%; spray voltage, 3.3 kV, capillary temperature, 320 °C; and heater temperature, 350 °C. The ions were scanned in a negative mode with a mass range from *m/z* 50 to 1000.

### 3.6. Evaluation of Antioxidant Activity

#### 3.6.1. DPPH Free Radical-Scavenging Assay

The DPPH radical-scavenging activity of the different forms of free, esterified, bound phenolic metabolites was measured according to a previously reported method [54]. The absorbance of the mixture was measured by a SpectraMax M5 microplate reader (Molecular Devices, San Jose, CA, USA) at 517 nm. The DPPH radical-scavenging activity (%) was calculated as [(A_control_ − A_sample_)/A_control_] × 100, where A_control_ and A_sample_ are the absorbance values for the incubation of the DPPH solution in absence and presence of the test compound.

#### 3.6.2. ABTS Radical-Scavenging Activity

The ABTS radical-scavenging activity of each extract was evaluated according to a previously reported method [55]. The absorbance of the mixture was measured with a SpectraMax M5 microplate reader at 745 nm. Similar to the DPPH antioxidant assay, the ABTS-scavenging activity (%) was calculated.

#### 3.6.3. Ferric-Reducing Antioxidant Power (FRAP) Assay

The FRAP assay was conducted by a previous method with some modifications [36]. Trolox FeSO_4_·7H_2_O was used as the standard to calibrate the standard curve, ranging from 0.1 to 0.5 mmol/L. The results were expressed as μmol Trolox/g extract.

### 3.7. Cytoprotective Effect of Different Phenolic Fractions in H_2_O_2_-Induced HepG2 Cells

#### 3.7.1. Cell Culture and Cytotoxic Assay

Human liver cancer cells HepG2 were purchased from the Kunming Cell Bank, Chinese Academy of Sciences (Kunming, China). The cells were cultured in DMEM with 10% FBS and 1% antibiotic mixture of penicillin (100 U/mL) and streptomycin (100 mg/mL) in a humidified atmosphere containing 5% CO_2_ and 95% air at 37 °C.

The cytotoxicity of the test samples on the HepG2 cells was determined by MTT assay with minor modifications [40]. Briefly, the cells were seeded at 1.0 × 10^5^ cells per well in 96-well culture plates. The cells were treated with or without different concentrations of the test sample and incubated for 24 h. Then, the cells were treated with 0.5 mg/mL MTT solution for 4 h. After removal of the MTT solution, 150 μL dimethyl sulfoxide (DMSO) was added in order to solubilize the purple formazan crystals. After being thoroughly dissolved, the absorbance was recorded at 570 nm by a microplate reader. The cell viability results demonstrated that all the samples were non-toxic to HepG2 cells at the tested concentrations.

#### 3.7.2. Inhibition of Different Phenolic Fractions on Reactive Oxygen Species (ROS) Generation in H_2_O_2_-Induced HepG2 Cells

The accumulation of intracellular total ROS in HepG2 cells was quantified by DCFH-DA staining, as described previously [56]. Briefly, the HepG2 cells were seeded in a 6-well plate at a density of 1 × 10^5^ cells per well for 24 h and exposed to the test sample at different concentrations of 0, 50, 100, and 200 μg/mL for another 24 h. The medium was removed, and the collected cells were washed with PBS and treated with 1.0 mM H_2_O_2_ for 6 h. Then, the cells were harvested and washed with cold PBS twice. The cells were incubated in serum-free medium with 10 mM DCFH-DA at 37 °C for 20 min in the dark. After incubation, the cells were washed twice using FBS-free medium and then immediately detected with flow cytometry (Guava^®^ easyCyte6-2 L, Millipore, Billerica, MA, USA).

#### 3.7.3. Spectrophotometric Determination of GSH and SOD Levels

The levels of superoxide dismutase (SOD) and glutathione (GSH) in the cultured cells were detected with a microplate reader using commercial kits (Sigma-Aldrich, Shanghai, China), following the manufacturer’s recommendations.

### 3.8. Enzyme Inhibitory Assay

#### 3.8.1. Inhibition Assay of α-Glycosidase Activity

The α-glucosidase inhibition assay was performed as previously described [57]. The test samples with different concentrations (50 mL) and α-glucosidase (100 μL) were incubated at 37 °C for 15 min. Then, 50 μL PNPG (3 mM) was added to the mixture and incubated for 15 min at 37 °C. Then, the reaction was terminated by adding 1 mL Na_2_CO_3_ (1 M). The absorbance of each reaction mixture was measured at 405 nm using a microplate reader. Acarbose was used as the positive control.

#### 3.8.2. Inhibition Assay of α-Amylase Enzyme Activity

The α-amylase inhibition assay of the PEFs was performed according to the previously reported assay with minor modifications [58]. Briefly, 50 μL of each sample at different concentrations was added into 50 μL α-amylase enzyme solution and 50 μL of a 1% starch solution. The reaction was stopped with 100 μL of dinitrosalicylic acid, reacted in a boiling water bath for 5 min and cooled to room temperature. The absorbance was recorded by a multi-detection microplate reader at 540 nm.

#### 3.8.3. Pancreatic Lipase Inhibition Assay

The lipase activity was determined according to previous work [59] using 4 mmol/L *p*-nitrophenyllaurate in Tris-HCl 0.05 mmol/L, pH 8.0 buffer containing 0.5% Triton-X100 as a substrate. The p-nitrophenol, a product of the lipase action on *p*-nitrophenyllaurate, was measured in a spectrophotometer at 410 nm with a microplate reader. Orlistat was used as the positive control.

### 3.9. Statistical Analysis

The experimental data were expressed as the mean ± standard deviation (SD). A one-way ANOVA and Tukey’s test were used to evaluate the significant differences. Statistical significance was defined as *p* < 0.05 for all the tests. All the analyses were conducted using the Origin 10.5 software (OriginLab, Northampton, MA, USA).

## 4. Conclusions

The free, esterified and bound phenolics were obtained from normal and UHP-treated PEFs by the acid and alkaline extraction method. UHP pretreatment significantly increased the TPC, TFC, and TTC values of the free, esterified, and bound phenolic fractions of PEFs compared to normal PEFs. A total of 24 phytoconstituents were identified in the PEFs. In addition, the UHP-treated PEFs exhibited better antioxidant activity, cell protective effect and enzyme inhibitory activity than the normal PEFs. Furthermore, PEFs have potential hypoglycemic activity via the inhibitory capacity on α-glycosidase. The hypoglycemic activity and mechanism can be further explored through in vivo and in vitro experiments to broaden the application range of PEFs. The above findings suggest that UHP treatment could be used to effectively enhance the extraction yield of phenolics and the bioactivities of different phenolic fractions obtained from PEFs. This research could promote the applications and the economic value of PEFs in functional food or the nutraceutical industry.

## Figures and Tables

**Figure 1 molecules-29-03181-f001:**
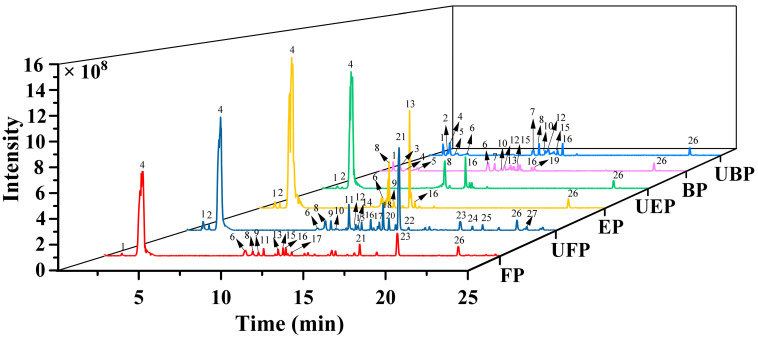
The total ion chromatograms of different phenolic fractions of PEFs in negative mode. FP, EP, and BP were obtained from normal PEFs. UFP, UEP, and UBP were obtained from UHP-treated PEFs.

**Figure 2 molecules-29-03181-f002:**
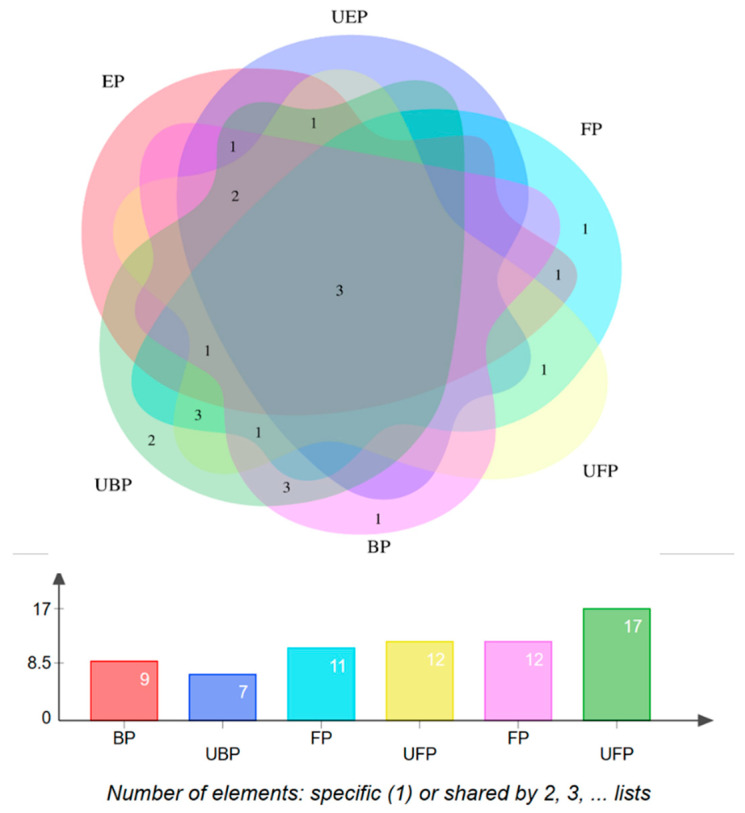
Chemical composition of the PEF distribution Venn diagram.

**Figure 3 molecules-29-03181-f003:**
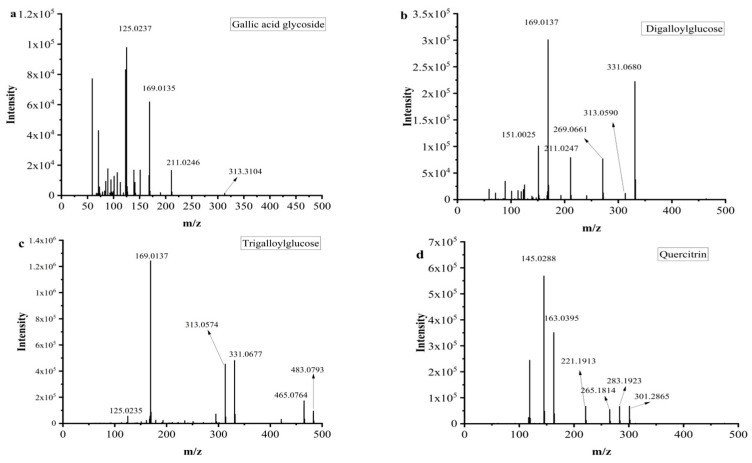
MS^2^ and possible fragmentation pathways of compounds **3**, **7**, **10**, **12** and **16**.

**Figure 4 molecules-29-03181-f004:**
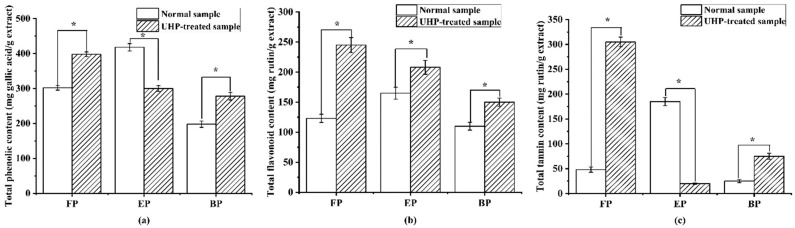
The content of total phenolics, flavonoids and tannins in the free, esterified and bound phenolic fractions of normal and ultra-high-pressure-treated *P. emblica* fruits: (**a**) total phenolic content; (**b**) total flavonoid content; and (**c**) total tannin content. * *p* < 0.05. FP, free phenolic fraction; EP, esterified phenolic fraction; BP, bound phenolic fraction.

**Figure 5 molecules-29-03181-f005:**
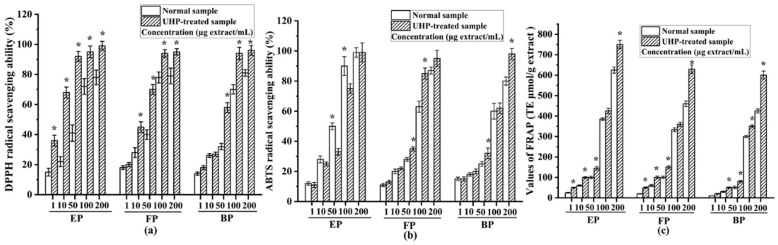
The antioxidant activities of the free, esterified and bound phenolic fractions from normal and UHP-treated PEFs. (**a**) The DPPH radical-scavenging activity. (**b**) The ABTS radical-scavenging activity. (**c**) The FRAP radical-scavenging activity. * Indicates significant differences between the same fractions from the normal and UHP-treated PEFs, respectively (*p* < 0.05). * *p* < 0.05. FP, free phenolic fraction; EP, esterified phenolic fraction; BP, bound phenolic fraction.

**Figure 6 molecules-29-03181-f006:**
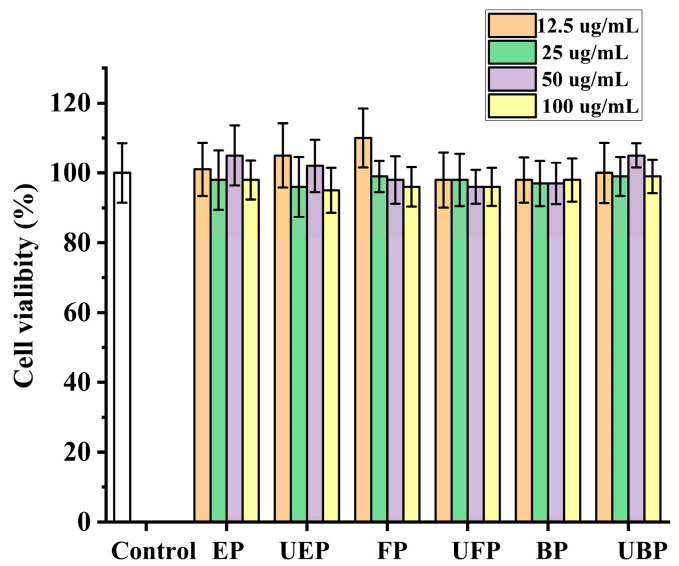
The effect of different fractions on HepG2 cell viability. FP, free phenolic fraction; EP, esterified phenolic fraction; BP, bound phenolic fraction. UBP, UHP-treated bound phenolic fraction; UFP, UHP-treated free phenolic fraction; UEP, UHP-treated esterified phenolic fraction.

**Figure 7 molecules-29-03181-f007:**
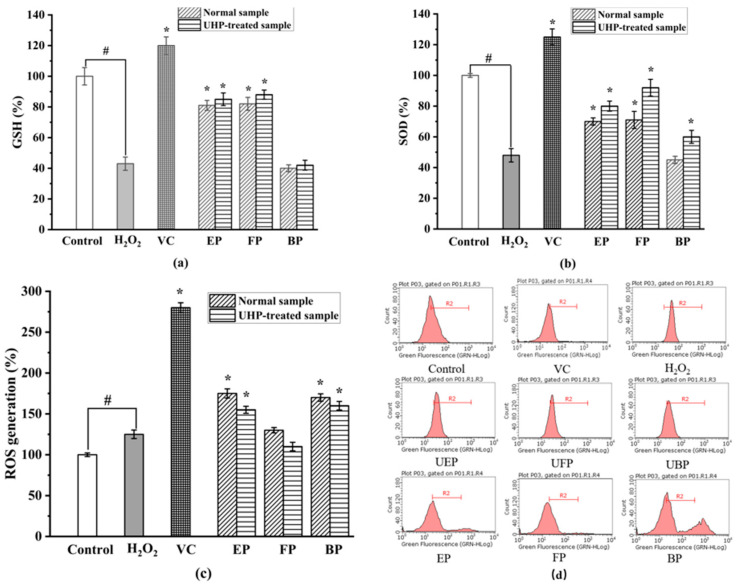
The effect on intracellular antioxidant enzyme activities. (**a**) Superoxide dismutase (SOD) activity. (**b**) Glutathione (GSH) activity. (**c**) Intracellular ROS inhibitory effects in H_2_O_2_-induced HepG2 cells. (**d**) The inhibitory effects of cellular ROS generation. FP, free phenolic fraction; EP, esterified phenolic fraction; BP, bound phenolic fraction. # control group vs. model group; * sample group vs. model group.

**Figure 8 molecules-29-03181-f008:**
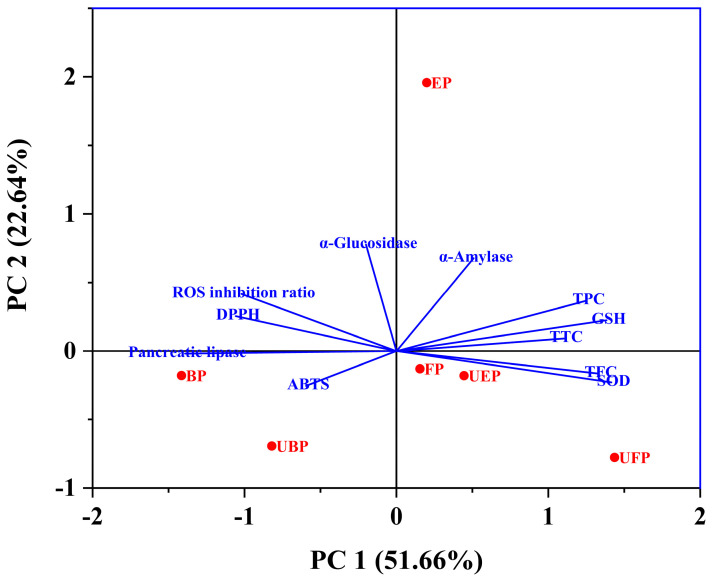
Principal component analysis (PCA) of the principal compounds, antioxidant, enzyme inhibitory and cytoprotective activities.

**Figure 9 molecules-29-03181-f009:**
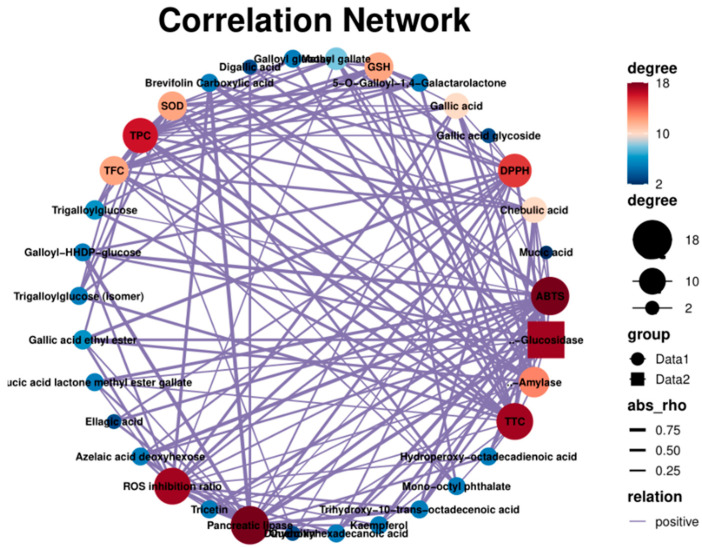
The correlation network of chemical constituents, antioxidant activity, enzyme activity and cytoprotective effects.

**Table 1 molecules-29-03181-t001:** Chemical profiling of *P. emblica* fruits by UHPLC-ESI-HRMS/MS in negative mode.

Peak	RT	[M-H]^-^ (*m/z*)	Molecular Formula	Error ppm	MS/MS	Compound	Reference
1	1.14	209.0302	C_6_H_8_O_7_	0.251	191.0126	Mucic acid	Standard
2	1.21	355.0317	C_12_H_22_O_11_	3.214	151.0394, 336.7523	Chebulic acid	Standard
3	1.76	331.0680	C_13_H_16_O_10_	6.183	169.0136	Gallic acid glycoside	[19]
4	2.41	169.0137	C_12_H_22_O_11_	5.332	125.0333	Gallic acid	Standard
5	3.19	343.0316	C_13_H_12_O_11_	5.838	123.0088, 169.0143	5-O-galloyl-1,4-galactarolactone	HMDB
6	8.92	183.0295	C_8_H_5_O_5_	6.754	169.0.24, 124.0153	Methyl gallate	Standard
7	9.39	483.0793	C_20_H_20_O_14_	2.163	151.0037, 169.0137, 331.0574	Digalloylglucose	[6]
8	9.42	321.0261	C_14_H_10_O_9_	0.458	125.0235, 169.0138	Digallic acid	Standard
9	9.78	291.0156	C_13_H_8_O_8_	1.768	191.0347, 219.0300, 247.0253	Brevifolin carboxylic acid	[20]
10	10.00	635.0914	C_27_H_24_O_18_	5.365	169.0137, 313.0574, 465.0690	Trigalloylglucose	[6]
11	10.10	633.0775	C_27_H_22_O_18_	6.423	301.0135, 463.1024	Galloyl-HHDP-glucose	[6,21]
12	10.26	635.0914	C_27_H_24_O_18_	3.431	169.0137, 313.0574, 465.0690	Trigalloylglucose (isomer)	[6]
13	10.96	197.0453	C_9_H_10_O_5_	6.258	69.0335, 125.0325, 162.8386	Gallic acid ethyl ester	[22]
14	11.07	357.0479	C_14_H_14_O_11_	3.925	169.0132	Mucic acid lactone methyl ester gallate	[6,23]
15	11.61	300.9998	C_14_H_6_O_8_	4.856	257.0099, 283.9971	Ellagic acid	Standard
16	11.45	447.0926	C_21_H_20_O_11_	5.894	283.1923, 301.2865	Quercitrin	[24,25]
17	11.89	187.0973	C_9_H_16_O_4_	4.553	125.0964, 169.0864	Azelaic acid	[26]
18	12.48	303.2186	C_15_H_10_O_7_	2.305	127.3785; 153.0151	Tricetin	[25]
19	12.67	399.0943	C_17_H_18_O_11_	5.443	125.0235, 169.0137	Unknown	-
20	14.13	301.2030	C_15_H_9_O_7_	4.594	125.0961, 243.0125	Quercetin	Standard
21	14.39	287.2235	C_16_H_32_O_4_	6.455	207.1757, 251.1664	Dihydroxyhexadecanoic acid	[27]
22	15.05	285.2080	C_15_H_10_O_6_	1.025	151.0031, 267.0293	Kaempferol	Standard
23	18.54	329.2343	C_18_H_34_O_5_	4.562	171.1018, 211.1348,	Trihydroxy-10-trans-octadecenoic acid	[28]
24	19.31	433.2608	C_25_H_38_O_6_	6.258	163.0395, 287.2235	Unknown	-
25	20.04	433.2607	C_25_H_38_O_6_	6.843	163.0395, 287.2237	Unknown	-
26	22.37	277.1452	C_16_H_22_O_4_	6.583	121.0284, 147.0082	Mono-octyl phthalate	[29]
27	23.02	311.2238	C_18_H_32_O_4_	6.268	183.0118	Hydroperoxy-octadecadienoic acid	[28]

**Table 2 molecules-29-03181-t002:** The IC_50_ value (mg/mL) of three different phenolic forms on the inhibitory activity of α-amylase, α-glycosidase and pancreatic lipase.

Enzyme	Positive Control ^&^	EP ^#^		FP ^#^		BP ^#^	
		N *	U *	N	U	N	U
Pancreatic lipase ^&^	46.43 ± 1.3 ^f^	241.25 ± 8.21 ^c^	214.17 ± 9.31 ^d^	62.92 ± 2.34 ^e^	48.73 ± 3.28 ^f^	613.89 ± 9.83 ^a^	439.34 ± 12.25 ^b^
α-glucosidase ^^^	0.89 ± 0.10 ^f^	312.4 ± 2.21 ^a^	19.72 ± 0.81 ^d^	19.68 ± 1.01 ^d^	1.74 ± 0.19 ^e^	87.65 ± 3.42 ^b^	36.82 ± 1.09 ^c^
α-amylase ^^^	2.41 ± 0.19 ^d^	6.74 ± 0.42 ^a^	5.82 ± 0.72 ^a^	4.32 ± 0.29 ^b^	4.09 ± 0.31 ^b^	3.89 ± 0.15 ^b^	3.48 ± 0.27 ^c^

^#^ EP, FP, and BP represent the esterified phenolic, free phenolic and bound phenolic fractions, respectively. * N, normal PEFs; U, UHP-treated PEFs. The results are expressed as the mean ± SD; different letters in the same column as superscripts are significantly different (*p* < 0.05). ^&^ Orlistat was used as the positive control. ^ Acarbose was used as the positive control.

## Data Availability

The original contributions presented in the study are included in the article/Appendix A, further inquiries can be directed to the corresponding authors.

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
