# Peer review of "Effects of Ultra-High-Pressure Treatment on Chemical Composition and Biological Activities of Free, Esterified and Bound Phenolics from Phyllanthus emblica L. Fruits"

_molecules, 2024, doi:10.3390/molecules29133181_

Round 1
Reviewer 1 Report
Comments and Suggestions for Authors
All my comments are in the attached file.

Comments on the Quality of English LanguageMinor
Author Response
- Provide list of abbreviations.
Response: Thank you for your suggestion and your comments are helpful to our content. We have added a list of abbreviations in the article.
Line 31-40: Abbreviations: UHP (Ultra-high pressure), PEF (Phyllanthus emblica Linn. Fruits), UHPLC-HRMS/MS (Ultra-high-performance liquid chromatography high-resolution mass spectrometry), TPC (Total phenolic content), TFC (Total flavonoid content), TTC (Total tannin content), DPPH (2,2-diphenyl-1-picryhydrazyl), ABTS (2,2′-azino-bis (3-ethylbenzothiazoline-6-sulfonic acid)), FRAP (Ferric reducing antioxidant power), ROS (Reactive oxygen species), MTT (Methylthiazol-2-yl-2,5-diphenyl tetrazolium bromide), SOD (Superoxide dismutase), GSH (glutathione), IC50 (Half maximal inhibitory concen-tration), FP (Free phenolic fraction), EP (Esterified phenolic fraction), BP (Bound phenolic fraction), UBP (UHP-treated bound phenolic fraction), UFP (UHP-treated free phenolic fraction), UEP (UHP treated esterified phenolic fraction).
- Provide some reference for introduction.
Response: We are grateful for the suggestion. We have modified the introduction part of the article and added references.
Line 62-65: Recently, many modern non-thermal techniques such as ultra-high-pressure homogenization, thin-film short wave ultraviolet radiation, high hydrostatic pressure have been established the improvement potential in the bioaccessibility and bioavailability of bound phenolics [12-14].
[12] Sauceda-Gálvez, J. N.; Roca-Couso, R.; Martinez-Garcia, M.; Hernández-Herrero, M. M.; Gervilla, R.; Roig-Sagués, A. X., Inactivation of ascospores of Talaromyces macrosporus and Neosartorya spinosa by UV-C, UHPH and their combination in clarified apple juice. Food Control 2019, 98, 120-125.
[13] Melhem, L. C. D.; Do Rosario, D. K. A.; Monteiro, M. L. G.; Conte, C. A., High-Pressure Processing and Natural Antimicrobials Combined Treatments on Bacterial Inactivation in Cured Meat. Sustainability 2022, 14, (17).
[14] Boasiako, T. A.; Boateng, I. D.; Ekumah, J. N.; Johnson, N. A. N.; Appiagyei, J.; Murtaza, M. S.; Mubeen, B.; Ma, Y. K., Advancing Sustainable Innovations in Mulberry Vinegar Production: A Critical Review on Non-Thermal Pre-Processing Technologies. Sustainability 2024, 16, (3).
- Talk about the need for process intensification methods (non-thermal)
Response: Thank you for your suggestion and your comments are helpful to our content. We have added a description of the necessity and advantages of non-heat treatment in the paper.
Line 62-65: Recently, many modern non-thermal techniques such as ultra-high-pressure homogenization, thin-film short wave ultraviolet radiation, high hydrostatic pressure has been established the improvement potential in the bioaccessibility and bioavailability of bound phenolics [12-14].
[14] Boasiako, T. A.; Boateng, I. D.; Ekumah, J. N.; Johnson, N. A. N.; Appiagyei, J.; Murtaza, M. S.; Mubeen, B.; Ma, Y. K., Advancing Sustainable Innovations in Mulberry Vinegar Production: A Critical Review on Non-Thermal Pre-Processing Technologies. Sustainability 2024, 16, (3).
- Run statistical analysis and incorporate it in the results and discussion section.
Response: We are grateful for the suggestion. We have conducted data analysis based on your recommendations.
Line 247-252: HepG2 cell viabilities were measured after the co-incubation with the different phenolic fractions from PEF with or without UHP pretreatment by me-thylthiazol-2-yl-2,5-diphenyl tetrazolium bromide (MTT) method [37]. As shown in Fig. 6, the cell viability after all samples treatment was greater than 90 %, and there was no significant difference to the control group. Hence, all of the samples were no toxic to HepG2 cells at the concentration of 100 μg/mL.
5 Provide latitude and longitude.
Response: We are grateful for the suggestion. We have added the latitude and longitude information.
Line 379-380: Fresh PEF was obtained from Pu’er city of Yunnan province of China in July 2017(22°83′42′′N, 100°99′05′′E).
- Dried to what moisture content?
Response: Thank you for your suggestion and your comments are helpful to our content. By measuring the quality difference of PEF before and after freeze-drying, we concluded that the moisture content after freeze-drying was 6-10 %.
- Why methanol and not ethanol?
Response: Thank you for your suggestion and your comments are helpful to our content.
Theoretically, since the polarity of methanol molecules is higher than that of ethanol molecules, the polar components in plant methanol extracts are more than those in ethanol extracts (the concentration of the same components is relatively high), and methanol shows a higher yield for phenolic components. In addition, the reason why we prefer methanol is very simple. Methanol is not azeotropic with water, and the boiling point is lower than that of ethanol, which is easy to remove. We provide some literature as evidence, as follows:
[1] Kennas A , Amellal-Chibane H .Comparison of five solvents in the extraction of phenolic antioxidants from pomegranate (Punica granatum L.) peel[J].The North African Journal of Food and Nutrition Research (NAJFNR), 2019(1).DOI:10.51745/NAJFNR.3.5.140-147.
[2] Boulekbache-Makhlouf, L.; Medouni, L.; Medouni-Adrar, S.; Arkoub, L.; Madani, K., Effect of solvents extraction on phenolic content and antioxidant activity of the byproduct of eggplant. Industrial Crops and Products 2013, 49, 668-674. DOI: 10.1016/j.indcrop.2013.06.009
[3] Seal T. Antioxidant activity of some wild edible plants of Meghalaya state of India: A comparison using two solvent extraction systems[J]. International Journal of Nutrition and Metabolism 2012, 4(3), 51-56. DOI: 10.5897/IJNAM11.060.
[4] A O Z , A H P , B S A I ,et al.Green and highly extraction of phenolic compounds and antioxidant capacity from kinkeliba (Combretum micranthum G. Don) by natural deep eutectic solvents (NADESs) using maceration, ultrasound-assisted extraction and homogenate-assisted extraction[J]. Arabian Journal of Chemistry, 2022, 15 ( 5), 103752. DOI: 10.1016/j.arabjc.2022.103752
- What time and temperature?
Response: Thank you for your suggestion and your comments are helpful to our content. We have added the information of centrifugation.
Line396-397: The supernatant was collected by centrifugation (1200 × g, 10 min, 25 ℃)
9 What temperature?
Response: Thank you for your suggestion and your comments are helpful to our content. We have increased the temperature information of rotary evaporation.
Line 397-398: and evaporated under vacuum using a rotary evaporator (45 ℃) to yield a crude extract (CE).
10 So, what was the internal standard? How did you ensure that there were no false positives?
Response: Thank you for your suggestion and your comments are helpful to our content. In quantitative analysis, internal standard is indispensable. In our experiment, we qualitatively analyzed the compounds contained in Phyllanthus emblica. Among them, Mucic acid, Chebulic acid, Methyl gallate, Digallic acid, Ellagic acid, Quercitin, Kaempferol and other compounds were compared with standard substances. Other compounds were compared for MS2 information using data reported in the literature, MASSBANK, and HMDB databases. In addition, in order to avoid the influence of solvent on the identification of compounds, we used solvent as blank control to avoid the influence of solvent peak.
11 Also pinpoint future studies that needs to be done.
Response: We are grateful for the suggestion. We have added research that can be carried out in the future to the article.
Line 556-560: Furthermore, PEF have potential hypoglycemic activity by the inhibitory capacity on α-glycosidase. The hypoglycemic activity and mechanism can be further explored through in vivo and in vitro experiments to broaden the application range of PEF.
Reviewer 2 Report
Comments and Suggestions for Authors
The authors showed that the pretreatment of raw Phyllanthus emblica Linn. fruits using ultra-high pressure has an effect on increasing the total content of phenols and flavonoids in the obtained extract, as well as that phenolic fractions have stronger antioxidant activities and an inhibitory effect on intracellular reactive oxygen species (ROS) and metabolic enzymes. A modern and sound methodology (UHP) was employed as a promising strategy for the improved extraction of bound phenols that are mainly bound to macromolecules in the cell wall. The methodology is generally well implemented and the results are well and clearly presented. In my opinion, manuscript can be published after some corrections:
1) To improve readability, it might be useful to change the order of the descriptions in the section Introduction: first about the PEF, then the active principles and at the end, about the extraction methods.
2) It would be clearer if MS2 and possible fragmentation pathways of the compounds presented were shown separately in Figure 3.
3) Some of the abbreviations (ABTS, TPTZ, DPPH) are explained in subsection 3.1. It would be more useful to explain them the first appear in the text.
4) Some English language corrections issues should be done through the text, (e.g, line 34, „various biological activities for improving human health“, line 53, „mainly distributed Southeast Asia“, line 382 „extracted three times 70% methanol“...)
Comments on the Quality of English LanguageSome English language corrections issues should be done through the text. Some of them are:
line 34, „various biological activities for improving human health“,
line 53, „mainly distributed Southeast Asia“,
line 382 „extracted three times 70% methanol“
Author Response
- To improve readability, it might be useful to change the order of the descriptions in the section Introduction: first about the PEF, then the active principles and at the end, about the extraction methods.
Response: Thank you for your suggestion and your comments are helpful to our content. According to your suggestion, we rewrite the introduction of the article.
Phyllanthus emblica L., belonging to the Euphorbiaceae family, is a peen-nial plants and is widely distributed in China, India, Thailand, Malaysia and other countries in Southeast Asia. It is an important economic crop in China. Its fruits have been widely processed as juice, beverage, jam, cosmetics due to its special taste, nutrients and bioactive phytochemicals. In addition, the fruits and roots had been traditionally used as herbal medicine for treating eczema, cough and sore throat in China. Many studied reported that P. emblica fruits (PEF) had significant anti-inflammatory, antibacterial, neuroprotective, hepa-toprotective, antihyperglycemic and antihyperlipidemic effects. Phytochemi-cal investigations on PEF reported a series of phenolic acids, flavonoids, and tannins in addition, these phenolic compounds are existed in bound form.
Phenolic compounds can be classified as free, esterified and bound forms based on their extractability and interaction with cell wall components. The free and esterified phenolics localized in the vacuoles of plant cells could be easily extracted by different organic solvents. The bound phenolics mostly bind to macromolecules in cell wall matrix such as cellulose, hemicellulose, lignin, pectin and proteins, thereby the extraction yield is low. Therefore, the research on the chemical composition and biological activity of bound phenolics is lim-ited [11]. In the past decade, how to improve the extract yield and illustrate the chemical constituents of bound phenolics had been attracted an increasing at-tention.
Recently, many modern non-thermal techniques such as ul-tra-high-pressure homogenization, thin-film short wave ultraviolet radiation, high hydrostatic pressure have been established the improvement potential in the bioaccessibility and bioavailability of bound phenolics [12-14]. Ultra-high hydrostatic pressure (UHP) they could damage the cell walls and the chemical hydrogen, ester, disulfide and carbon-carbon bonds between bound compo-nents and plant cell matrix. Some studies had been evidenced that UHP treat-ment could increase the extraction yield of polysaccharide [15], flavonoids [16] and carotenoid [17]. in a short time at low temperature [18].
In this study, the UHP processing method were used to extract the free, esterified, and bound phenolics in PEF. Furthermore, the chemical composition and their antioxidant activities, cytoprotective effects and enzyme inhibitory activity of different phenolic fractions were determined.
- It would be clearer if MS2 and possible fragmentation pathways of the compounds presented were shown separately in Figure 3.
Thank you for your suggestion and your comments are helpful to our content. We have adjusted MS2 and possible fragmentation pathways of the compounds.
- Some of the abbreviations (ABTS, TPTZ, DPPH) are explained in subsection 3.1. It would be more useful to explain them the first appear in the text.
Response: Thank you for your suggestion and your comments are helpful to our content. We have added acronyms to the article and explained them when they first appeared in the article.
Line 31-40: Abbreviations: UHP (Ultra-high pressure), PEF (Phyllanthus emblica Linn. Fruits), UHPLC-HRMS/MS (Ultra-high-performance liquid chromatography high-resolution mass spectrometry), TPC (Total phenolic content), TFC (Total flavonoid content), TTC (Total tannin content), DPPH (2,2-diphenyl-1-picryhydrazyl), ABTS (2,2′-azino-bis (3-ethylbenzothiazoline-6-sulfonic acid)), FRAP (Ferric reducing antioxidant power), ROS (Reactive oxygen species), MTT (Methylthiazol-2-yl-2,5-diphenyl tetrazolium bromide), SOD (Superoxide dismutase), GSH (glutathione), IC50 (Half maximal inhibitory concen-tration), FP (Free phenolic fraction), EP (Esterified phenolic fraction), BP (Bound phenolic fraction), UBP (UHP-treated bound phenolic fraction), UFP (UHP-treated free phenolic fraction), UEP (UHP treated esterified phenolic fraction).
Line 86-88: Phenolic metabolites in free, esterified and bound forms from PEF were analyzed by an Ultra-high-performance liquid chromatography high-resolution mass spectrometry (UHPLC-HRMS/MS).
Line 172-173: Among the three phenolic fractions of normal PEF, the esterified phenolic frac-tion (EP).
Line 175-176: followed by free phenolic fraction (FP). The bound phenolic fraction (BP) had the lowest TPC
Line 194-199: In this study, the antioxidant activities of all phenolic fractions from nor-mal and UHP-treated PEF were evaluated in terms of the scavenging abilities of 2,2-diphenyl-1-picryhydrazyl (DPPH), (2,2′-azino-bis (3-ethylbenzothiazoline-6-sulfonic acid)) (ABTS), Ferric reducing antioxidant power (FRAP), the inhibitory effect on intracellular reactive oxygen species (ROS) generation and intracellular antioxidant enzyme activities.
Line 204-205: The Half maximal inhibitory concentration (IC50) values of the esterified
Line 256-258: HepG2 cell viabilities were measured after the co-incubation with the dif-ferent phenolic fractions with or without UHP pretreatment by methylthiazol-2-yl-2,5-diphenyl tetrazolium bromide (MTT) method.
Line 269-271: Endogenous antioxidant enzyme systems including superoxide dismutase (SOD) and glutathione (GSH) played an indispensable role in preventing oxidative stress.
- Some English language corrections issues should be done through the text, (e.g, line 34, „various biological activities for improving human health“, line 53, „mainly distributed Southeast Asia“, line 382 „extracted three times 70% methanol“...).
Response: Thank you for your suggestion and your comments are helpful to our content. We have modified the English expression in the article.
Line 42-44: Phyllanthus emblica L., belonging to the Euphorbiaceae family, is a peren-nial plant and is widely distributed in China, India, Thailand, Malaysia and other countries in Southeast Asia.
Line 242-245: Phenolics could protect cells from oxidative damage by inhibiting the production of excessive ROS [42]. In this study, whether the different phenolic fractions of PEF had the capacity to inhibit ROS generation were performed in the H2O2-induced HepG2 cells.
Line 275-279: There was a significant difference between the SOD activity of normal and UHP-treated PEF. Among the three different phenolic fractions from normal and UHP-treated PEF, FP exhibited the strongest improvement activity on SOD expression. By comparison with the control group, the FP increased by 49%, followed by the EP (p < 0.05).
Line 292-305: As shown in Fig .7d, the relative amount of intracellular ROS significantly increased to 212.33 ± 10.67% after the H2O2 treatment, when compared with the control group. After the treatment of six fractions from the PEF with or without UHP pretreatment, the ROS production effectively decreased in a concentration-dependent manner (p < 0.05). The FP, EP, and BP from UHP-treated PEF had the lower values than those from normal PEF (p < 0.05), indicating that UHP-treated could effectively enhance the inhibitory effect of three phenolic fractions on intracellular ROS generation. Additionally, the FP exhibited the strongest scavenging activity of intracellular ROS, whereas the BP showed the weakest intracellular ROS inhibitory effect in comparison with other extracts (Fig .7c). The inhibitory effect of the three phenolic components on intracellular ROS production might be related to the chemical consituents and contents of phenolics, flavonoids and tannins [43]. In addtion, the syner-gistic effects between different phenolic compounds might be another influ-ence factor [42].
Line 308-311: The inhibitory effects of these phenolic fractions againstα-glucosidase and α-amylase were determined and the results were described in Table 2. All phenolic fractions showed significant inhibitory capacity on α-glucosidase and α-amylase (p < 0.05).
Line 393-396: After filtration with filter paper, the degreased sample was obtained and extracted by 70% methanol at the ratio of mate-rial to liquid 1:10 for 30 min by ultrasound-assisted extraction (n=3).
Round 2
Reviewer 1 Report
Comments and Suggestions for Authors
accept
Comments on the Quality of English Languageminor